# Management of Intrahepatic Cholangiocarcinoma: A Narrative Review

**DOI:** 10.3390/cancers16040739

**Published:** 2024-02-10

**Authors:** Carolyn Tsung, Patrick L. Quinn, Aslam Ejaz

**Affiliations:** 1Department of Surgery, The Ohio State University Wexner Medical Center, Columbus, OH 43210, USA; catsung77@gmail.com (C.T.); patrick.quinn@osumc.edu (P.L.Q.); 2Department of Surgery, University of Illinois at Chicago, Chicago, IL 60612, USA

**Keywords:** intrahepatic cholangiocarcinoma, adjuvant, systemic therapy, immunotherapy, targeted therapy, precision medicine

## Abstract

**Simple Summary:**

Intrahepatic cholangiocarcinoma is a rare primary liver tumor with a poor prognosis and a 5-year survival rate of 9%. While surgical resection offers the best chance of long-term survival, only a subset of patients will present with resectable disease at presentation, and upwards of 70% of patients experience recurrence following resection. Consequently, systemic therapies play an important role in the management of resectable disease. This review summarizes the approach to resectable intrahepatic cholangiocarcinoma and reviews current evidence from clinical trials regarding systemic and targeted therapies.

**Abstract:**

The management of resectable intrahepatic cholangiocarcinoma remains a challenge due to the high risk of recurrence. Numerous clinical trials have identified effective systemic therapies for advanced biliary tract cancer; however, fewer trials have evaluated systemic therapies in the perioperative period. The objective of this review is to summarize the current recommendations regarding the diagnosis, surgical resection, and systemic therapy for anatomically resectable intrahepatic cholangiocarcinoma. Our review demonstrates that surgical resection with microscopic negative margins and lymphadenectomy remains the cornerstone of treatment. High-level evidence regarding specific systemic therapies for use in resectable intrahepatic cholangiocarcinoma remains sparse, as most of the evidence is extrapolated from trials involving heterogeneous tumor populations. Targeted therapies are an evolving practice for intrahepatic cholangiocarcinoma with most evidence coming from phase II trials. Future research is required to evaluate the use of neoadjuvant therapy for patients with resectable and borderline resectable disease.

## 1. Introduction

Cholangiocarcinoma is a rare and aggressive collection of tumors that arise from the epithelial cells of bile ducts. The incidence within the United States and globally is rising with cholangiocarcinoma currently estimated to account for 3% of all gastrointestinal malignancies and 15% of all primary liver cancers [1,2]. Given the often late presentation of cholangiocarcinoma, intrahepatic cholangiocarcinoma carries a dismal 5-year survival rate of 9% [3]. Surgical resection offers the best chance of a cure for those with technically resectable disease with reported 5-year survival rates up to 40% following resection [4,5]. However, only a subset of patients present with resectable disease, and many patients who undergo resection recur within two years [6,7]. This highlights the need for more effective systemic therapies as an adjunct to surgical resection to improve long-term outcomes.

Found anywhere along the biliary tree, cholangiocarcinoma is categorized into subtypes based on location: intrahepatic (ICC), perihilar, and distal. Approximately 10% of cholangiocarcinomas are intrahepatic, defined as tumors arising proximal to second-order bile ducts within the hepatic parenchyma [8]. In conjunction with different anatomical locations, intrahepatic and extrahepatic cholangiocarcinomas are biologically different tumors, resulting in different clinical features and management [9]. With genetic sequencing, recent advances have been made in understanding the genetic underpinnings specific to ICC. These advances have led to the possibility of precision medicine, targeted therapies specific to a tumor’s genetic makeup, to be used for the management of ICC [10].

While surgical resection remains the cornerstone of treatment for ICC, multimodal therapy with systemic and novel targeted agents is likely needed to optimize outcomes. The purpose of our review is to examine the evidence regarding the diagnosis and treatment of ICC and to evaluate the utility of systemic and targeted therapies in the perioperative period.

## 2. Materials and Methods

For this narrative review, a literature search was performed to identify clinical trials involving neoadjuvant or adjuvant therapy for patients with ICC undergoing resection using MEDLINE/PubMed with an end search date of 24 July 2023. The search strategy involved using the terms “intrahepatic cholangiocarcinoma”, “bile duct cancer”, “resection”, “surgery”, “systemic therapy”, “chemotherapy”, “targeted therapy”, “immunotherapy”, and “clinical trials”. Non-English articles were excluded. Studies focusing on locoregional therapies and transplantation for ICC were also excluded. A comprehensive review of the eligible literature was performed by the authors with the most relevant, up-to-date articles included.

## 3. Discussion

### 3.1. Preoperative Evaluation

Given the involvement of the proximal bile ducts, ICC does not often present with jaundice as is observed more commonly with perihilar and distal cholangiocarcinoma [11]. Symptoms are often vague, such as nausea, bloating, abdominal discomfort, or weight loss. Early-stage tumors are frequently asymptomatic and are commonly found incidentally on cross-sectional imaging performed for other clinical reasons. Due to the lack of biliary obstruction, liver function tests are also non-specific in the setting of ICC. As a result, a high clinical suspicion is required for diagnosis in the early stage.

If suspected, a thorough history and physical examination should be the first step in the evaluation with a focus on risk factors for chronic liver inflammation (e.g., primary sclerosing cholangitis, metabolic dysfunction-associated steatotic liver disease, alcoholic cirrhosis, viral hepatitis, autoimmune hepatitis), hepatotoxin exposure (thoratrast, aflatoxin), travel history (liver fluke infection), family history of liver malignancy, and personal colorectal cancer screening history. In addition to a routine laboratory workup, including liver function tests, the tumor markers carcinoembryonic antigen (CEA), carbohydrate antigen 19-9 (CA 19-9), and alpha-fetoprotein (AFP) should be included. While CA 19-9 is non-specific, given elevations in inflammatory states as well as other cancers, it has been demonstrated to have a 72% sensitivity rate for ICC [12]. However, it should be noted that 5–10% of the Caucasian population does not synthesize CA 19-9 and is at risk for false negatives [13].

Imaging is crucial for the diagnosis and assessment of resectability with both multiphasic computed tomography (CT) or magnetic resonance imaging (MRI) with intravenous contrast as effective modalities. ICC can be distinguished from HCC, as there is typically progressive contrast uptake in both arterial and venous phases of cross-sectional imaging, whereas HCC is more classically associated with washout during the venous phase [14]. Additional image findings that suggest ICC include liver capsule retraction and biliary dilation in the vicinity of the lesion. It is important to note that a third of patients may not have these classic findings on imaging [15]. A CT chest should be obtained as part of the cancer staging. An image-guided biopsy can be a useful adjunct if imaging is equivocal. If a biopsy reveals adenocarcinoma, an esophagogastroduodenoscopy (EGD) and colonoscopy should be performed to rule out a metastatic liver lesion. Comprehensive molecular profiling is recommended for patients with unresectable or metastatic disease who are candidates for systemic therapy [16].

At present, there are no evidence-based regimens for neoadjuvant systemic therapy with guidelines recommending that resectable patients proceed directly to surgery (Table 1). To date, there has been only one prospective trial published that evaluated the safety and feasibility of neoadjuvant chemotherapy for patients with resectable ICC. In this multi-institutional phase II trial, the authors evaluated the GAP regimen (gemcitabine, cisplatin, and nab-paclitaxel) and found that it is both safe and feasible in the neoadjuvant setting without adversely impacting perioperative outcomes [17]. In this trial, patients were identified as high-risk based on various tumor-specific factors (Table 1). The authors found that the neoadjuvant GAP regimen was both safe and feasible without adversely impacting perioperative outcomes. As such, it is reasonable to consider that patients at high-risk for recurrence receive neoadjuvant therapy, preferably as a participant in a clinical trial.

### 3.2. Surgical Management

Surgical resection is currently the only potentially curative treatment for patients with ICC. In addition to the initial diagnostic work-up, further pre-operative evaluation includes an assessment of the patient’s medical comorbidities and functional status, quality of the patient’s liver function, anatomic extent of disease, and technical feasibility of resection. The goal of resection is to achieve negative hepatic margins and perform adequate lymph node sampling while preserving sufficient liver function with an adequately sized future liver remnant (FLR). For healthy individuals without any liver disease, an FLR of 20–25% is acceptable; for those with hepatic steatosis or chemotherapy-associated liver injury, an FLR of 30% is needed, whereas those with severe liver disease require an FLR of at least 40–50% [18]. As such, the resectability of the tumor depends on several factors, including its location within the liver, its relationship to major vessels and bile ducts, lymph node involvement, and the presence or absence of multifocal disease, satellite lesions, or metastases [19]. Provided that up to a third of patients may have occult metastatic disease, diagnostic laparoscopy should be considered as a staging tool at the time of curative-intent surgery to avoid futile laparotomy [20]. A recent systematic review of diagnostic laparoscopy in patients with ICC found that the yield may still be as high as 20%; however, the review was unable to identify any risk factors and, thus, concluded that the criteria and indications remain unclear.

Surgical resection can be approached via a traditional open approach or a minimally invasive approach by surgeons with technical expertise in this area. In small retrospective studies with appropriate patient selection, laparoscopic or robotic hepatic resection has demonstrated equivalent oncologic outcomes with potentially less blood loss, decreased postoperative pain, fewer postoperative complications, and faster recovery times [21]. As stated above, curative-intent resection requires a microscopic (R0) negative margin, as positive resection margins have been demonstrated to be a poor prognostic factor for long-term outcomes along with multifocal disease, vascular invasion, and lymph node metastases [22]. Additionally, margin width has been independently associated with recurrence-free and overall survival with margins less than one centimeter associated with poorer survival as compared to larger margins [23]. A meta-analysis that included a total of 3007 patients from eleven studies found that achieving a margin width of more than five or ten millimeters was significant in improving survival [24]. While parenchymal-sparing resections have been well-established in the surgical treatment of HCC and colorectal liver metastases, the impact of nonanatomic resection for ICC on long-term outcomes remains debated [25,26]. At present, the emphasis remains on obtaining an R0 margin, not on the extent of hepatic resection. Major vascular resection may be required to achieve such margins and can be performed with similar perioperative and oncologic outcomes compared to those who do not require vascular resection [27,28].

Lymph node status is another important prognostic factor and helps guide adjuvant therapy. As a result, regional lymphadenectomy is routinely recommended as part of hepatic resection for ICC with the National Comprehensive Cancer Network (NCCN) recommending the removal of at least six nodes. A standard porta hepatis lymphadenectomy should include the resection of lymph nodes within the hepatoduodenal ligament (station 12) and along the common hepatic artery (station 8). Further consideration should be given to the tumor location and lymphatic draining patterns, as lymph nodes along the lesser curvature of the stomach and those within the retropancreatic region may need to be included in addition to the standard lymphadenectomy, depending on the tumor location [29]. While lymphadenectomy provides adequate staging and prognostication, it has not been demonstrated to have an overall survival benefit with one meta-analysis of seventeen studies demonstrating no significant difference in disease-free and overall survival between patients undergoing lymph node dissection and those who did not [30]. Taken together, lymphadenectomy helps to identify node-positive patients at high-risk for recurrence who may benefit from systemic adjuvant therapy as described in the BILCAP trial.

### 3.3. Adjuvant Therapy

While surgical resection remains the foundation of curative therapy for ICC, the high recurrence rates demonstrate the need for effective adjuvant systemic therapies to eradicate micro-metastatic disease. In a meta-analysis of twenty studies, including one randomized controlled trial, adjuvant chemotherapy and chemoradiation were demonstrated to improve overall survival for patients with resected biliary tract cancers (BTC), particularly for those with nodal disease or positive margins [31]. The current evidence supporting adjuvant therapy has come primarily from retrospective studies due to the low incidence of ICC and difficulties with accrual for randomized controlled trials. Furthermore, these studies often combine patients with gallbladder cancer and cholangiocarcinoma, which can make their results difficult to interpret given the distinct biological nature of these tumors. Below, we highlight several randomized controlled trials that evaluated adjuvant therapy for ICC.

The NCCN recommends capecitabine as the first-line adjuvant therapy following resection for ICC. This is based on the results from the BILCAP trial, a phase III multicenter, randomized controlled trial performed in the United Kingdom from 2006 to 2014 [32]. This trial included 447 patients with biliary tract or gallbladder cancer (19% with ICC) who were randomized to capecitabine for eight cycles or observation following R0 or R1 resection. Among the intention-to-treat patients, 38% of patients underwent an R1 resection, and 47% of patients had nodal disease. There was no significant difference in overall survival in the intention-to-treat analysis (hazard ratio [HR] 0.81, 95% confidence interval [CI] 0.63–1.04; *p* = 0.097); however, in the per-protocol analysis of 430 patients, the median overall survival was 53 months in the capecitabine group and 36 months in the observation group (adjusted HR 0.75, 95% CI 0.58–0.97; *p* = 0.028). These results were upheld in a long-term analysis of the BILCAP data with a median follow-up time of 106 months [33]. ACTICCA-1 is an ongoing phase III multicenter, randomized controlled trial examining adjuvant gemcitabine and cisplatin versus capecitabine for patients with resected BTC [34].

Two phase III randomized controlled trials examining adjuvant therapy for resected BTCs have been completed in Japan. The first, completed by Takada et al. from 1986 to 1992, randomized patients with resected pancreaticobiliary carcinomas to adjuvant mitomycin C and 5-fluorouracil or surveillance [35]. The study enrolled 508 patients of which 139 (27.4%) had BTC without specification of location within the biliary tree. Of the enrolled BTC patients, 118 were eligible for final analysis with 103 (87.2%) having nodal disease and 46 (39%) undergoing an R1 resection. There was no significant difference in 5-year disease-free survival between the intervention arm and control group for patients with BTC (20.7% vs. 15.0%, *p* = 0.89). However, a sub-analysis did demonstrate a survival benefit for patients with gallbladder carcinoma. The second randomized trial is the ASCOT trial, where patients with BTC who underwent R0/R1 resection between 2013 and 2018 were randomized to adjuvant S-1, an oral fluoropyrimidine derivative, or observation [36]. There were 440 patients enrolled of which 58 (13.2%) had ICC, 176 (40%) had nodal disease, and 64 (14.5%) underwent an R1 resection. The intervention arm demonstrated improved overall 3-year survival (adjusted HR 0.69; *p* = 0.008) and 3-year relapse-free survival (RFS) (HR 0.80, *p* = 0.088) for S-1 among the entire cohort.

The PRODIGE 12-ACCORD 18 trial, a phase III multicenter, randomized controlled trial completed in France, assessed RFS between patients with R0/R1 resected BTC randomized to gemcitabine and oxaliplatin or surveillance alone [37]. Of the 196 included patients, 43.9% had ICC, 87% had R0 resections, and 36% had nodal disease. There was no significant difference in the median RFS (30.4 months vs. 18.5 months, *p* = 0.48) or overall survival (75.8 months vs. 50.8 months, *p* = 0.74) between the intervention arm and control group.

There are limited clinical trial data regarding the use of adjuvant radiation therapy for BTCs. The current evidence to support its use, particularly in residual disease and nodal metastasis cases, arises from retrospective analyses [31]. The NCCN lists fluoropyrimidine-based chemoradiation as a possible management option for R1 resection or nodal-positive ICC based on data extrapolated from the SWOG S0809 study. This study was a phase II trial that examined adjuvant gemcitabine–capecitabine, followed by concurrent capecitabine and radiotherapy, for patients with R0/R1 resected extrahepatic cholangiocarcinoma or gallbladder carcinoma [38]. A total of 79 patients were enrolled; a total of 86% completed the regimen with a median overall survival of 35 months (R0 34, R1 35 months). Further exploration is needed in the form of phase III trials.

### 3.4. Systemic Therapies for Disease Recurrence

Recurrent disease is defined as the return of ICC six months after curative-intent resection or completion of adjuvant therapy. For patients who experience disease recurrence, the first-line therapy recommendation is gemcitabine, cisplatin, and duravalumab. This is based on the results of the recently reported TOPAZ-1 trial, a phase III international multicenter trial that compared gemcitabine plus cisplatin with either durvalumab or placebo, followed by durvalumab or placebo monotherapy, for patients with previously untreated advanced biliary tract cancer or those with recurrent disease [39]. The study included 685 patients: a total of 55.9% with ICC with 19.1% of the cohort having recurrent disease. The combination of immunotherapy and chemotherapy demonstrated improved survival with an HR for overall survival of 0.80 (*p* = 0.021).

The combination of cisplatin and gemcitabine for advanced BTC has been well-established for over ten years, based on the findings of the ABC-02 trial [40]. This phase III trial randomized patients with advanced BTC to receive gemcitabine plus cisplatin or gemcitabine alone. The trial included 410 patients who had a median follow-up of 8.2 months. The combination therapy demonstrated a significant survival benefit over gemcitabine monotherapy with the intervention arm having an overall survival of 11.7 months in comparison to 8.1 months among the monotherapy group (HR 0.64; *p* < 0.001). A phase II trial examined the addition of nab-paclitaxel to gemcitabine and cisplatin (GAP regimen) for 60 patients with advanced BTC, finding this combination to be safe with a median overall survival of 19.2 months [41]. Further investigation via a phase III trial is underway [42].

The recommendation for second-line therapy for advanced or recurrent ICC is folinic acid, fluorouracil, and oxaliplatin (FOLFOX). This is supported by the results of the ABC-06 trial, a phase III multicenter trial completed in the United Kingdom, which randomized 162 patients with advanced BTC with disease progression on first-line therapy to FOLFOX or supportive care [43]. FOLFOX was found to improve overall survival with a median overall survival of 6.2 months in comparison to 5.3 months for the control arm (HR 0.69; *p* = 0.031). Other second-line regimens, such as regorafenib or liposomal irinotecan, fluorouracil, and leucovorin, have demonstrated safety and efficacy in phase II trials [44,45].

### 3.5. Targeted Therapies for Disease Recurrence

Provided that several clinically relevant molecular alterations have been identified in ICC, several studies have evaluated the use of targeted therapies, particularly for advanced disease. In the most recent iteration of the NCCN guidelines, molecular testing is now recommended for patients with unresectable or metastatic BTC [16]. We recommend additional consideration for genetic testing at the time of resection, given the high prevalence of disease recurrence. The targets of interest include isocitrate dehydrogenase-1 (IDH1), fibroblast growth factor receptor-2 (FGFR2), human epidermal growth factor receptor-2 (HER2/ERBB2), protooncogene B-Raf (BRAF), high tumor mutational burden (TMB-H), high microsatellite instability or mismatch repair deficient (MSI-H/dMMR), neurotrophic tyrosine receptor kinase (NTRK) fusions, and RET fusions (Table 2). Investigations into these novel agents are ongoing with several receiving approval for use in advanced ICC from the Food and Drug Administration (FDA).

The two mutations found more frequently among ICC than the other BTCs are IDH1 mutations and FGFR2 fusions or rearrangements. Approximately 10–20% of ICC has IDH1 mutations and can be targeted with ivosidenib [46]. The ClarIDHy phase III multicenter trial randomized 185 patients with IDH1-mutated cholangiocarcinoma (91.4% ICC) that had progressed on standard chemotherapy to either ivosidenib or placebo [47]. After a median follow-up of 6.9 months, ivosidenib was found to improve progression-free survival (PFS) over the placebo with a median PFS of 2.7 months versus 1.4 months (HR 0.37; one-sided *p* < 0.0001). Adjusting for crossover between arms, the median overall survival was 10.3 months with ivosidenib compared to 5.1 months for the placebo (HR 0.49; one-sided *p* < 0.001) [48]. The 9–15% of FGFR2 mutations identified in ICC can be targeted with futibatinib, pemigatinib, and infigratinib [46]. At present, the evidence is limited to phase II trials examining these agents in previously treated advanced cholangiocarcinoma; however, the results have been promising to suggest a clinical benefit without a compromising adverse-effect profile [49,50,51].

Though found more prevalent in gallbladder carcinoma, an estimated 5–20% of cholangiocarcinomas harbor HER2 amplification or overexpression [52]. Early phase II trials did not demonstrate the efficacy of lapatinib—a dual tyrosine kinase inhibitor that interrupts the HER2/neu and EGFR pathways. However, the criteria for enrollment for these studies did not require HER2 amplification or overexpression [53,54]. More recent phase II trials have demonstrated improved efficacy with HER2-targeted therapies, such as the combination of trastuzumab and pertuzumab, which yielded a 23% objective response rate (ORR) in patients with metastatic BTC with overexpression or amplification of HER2 [55].

The remainder of the aforementioned targets are estimated to be present in less than 5% of BTCs [56]. BRAF V600E mutations have been targeted with the regimen of dabrafenib and trametinib. This oral combination was evaluated in the phase II ROAR basket trial with a subset of 43 advanced cholangiocarcinoma patients with previous systemic therapy and BRAF V600E mutations, demonstrating an overall response rate of 51% [57]. The current recommendations for immunotherapy come from two phase II trials. While better established for the treatment of advanced melanoma and non-small cell lung cancer, the combination of nivolumab and ipilimumab has been evaluated for other solid tumors with TMB-H in the CheckMate 848 study. Preliminary results have demonstrated that, within the 68 patients with tissue TMB-H, the ORR was 35.3% (95% CI 24.1–47.85) [58]. In the KEYNOTE-158 cohort K study, 321 patients with previously treated non-colorectal MSI-H/dMMR solid tumors, including 22 patients with cholangiocarcinoma, were treated with pembrolizumab [59]. The ORR for the entire cohort was 30.8% with analysis of the cholangiocarcinoma group demonstrating an ORR of 40.9% (95% CI 20.7–63.6%).

The last two targets, RET and NTRK fusions, are rare within BTC and present in less than 1% of patients. Entrectinib and larotrectinib are approved for use in patients with NTRK fusions. Several phase I/II basket trials demonstrated ORRs over 50%; however, it should be noted that only three patients with cholangiocarcinoma were included in these studies [60,61]. Similar evidence exists for the RET inhibitors selpercatinib and pralsetinib, which each demonstrated efficacy above 40% in basket phase I/II trials. However, each study had three or fewer patients with cholangiocarcinoma [62,63]. This further highlights the difficulty of establishing high-level evidence for the use of target therapies within ICC due to challenges with low accrual. Improvements continue to be made with next-generation sequencing, and additional targets for ICC are expected to be identified in the future.

## 4. Conclusions and Future Directions

ICC is a rare biliary tract cancer that typically presents at a late stage as a result of its indolent course. Surgical resection with negative microscopic margins and porta hepatis lymphadenectomy provides the best chance for long-term survival. Even with curative-intent resection, most patients are still at high risk for tumor recurrence. We suggest that molecular testing be performed on the resected tumor to help aid in the selection of adjuvant therapy, particularly for those with the highest risk for recurrence, including positive margin status and lymph node-positive disease among others. For all patients with adequate performance status, adjuvant chemotherapy with capecitabine should be pursued as a first-line agent. In cases of disease recurrence or progression, the combination of durvalumab, gemcitabine, and cisplatin should be considered as a first-line regimen. Targeted therapies are also a possible option, pending the results of genetic testing, with an individualized approach for each patient.

Promising ongoing clinical trials for patients with potentially resectable ICC are largely focused on perioperative administration of immune-based, cytotoxic, or molecular-based personalized targeted therapies in patients undergoing curative-intent resection. Ongoing adjuvant trials following curative-intent resection of ICC include the ATTICA-1 trial (phase 3 randomized trial of gemcitabine/cisplatin vs. observation), the GAIN trial (phase 3 randomized trial of perioperative gemcitabine/cisplatin vs. surgery alone), and the OPTIC trial (phase II trial of neoadjuvant nab-paclitaxel, cisplatin, and gemcitabine with or without infigratinib).

While this review presents a broad overview of the current recommendations for the systemic therapies in use for resectable ICC, it should be noted that this review is not exhaustive, and not every potential agent is included. The field of precision medicine is rapidly changing with a plethora of ongoing clinical trials identifying new novel agents and increasing our understanding of the benefits of therapies currently in use. Provided this swift accumulation of new evidence, clinicians must monitor ongoing evidence-based recommendations made at the national level. Additionally, it should also be acknowledged that many of the targeted therapies currently in use are based on findings from phase II trials with heterogenous primary tumors with only ivosidenib having demonstrated a survival benefit in a phase III trial. The decision to use such targeted therapies should be made after a multidisciplinary tumor board discussion.

Although the incidence of cholangiocarcinoma is on the rise, ICC is still a rare tumor, resulting in difficulties in accrual for randomized controlled trials. Multicenter collaboration at the national and international levels will be imperative to adequately power trials for high-level evidence. Future studies aimed at studying the impact of the aforementioned therapies as well as assessing neoadjuvant treatment are needed.

## Figures and Tables

**Table 1 cancers-16-00739-t001:** High-risk ICC tumors as defined in the NEO-GAP trial.

High-Risk ICC Factors
T-Stage 1b or greater
Solitary lesion greater than 5 cm
Multifocal tumors
Satellite lesions confined to the same lobe of the liver as the dominant lesion and technically resectable
Presence of major vascular invasion but technically resectable
Suspicious or involved regional lymph nodes (N1)

**Table 2 cancers-16-00739-t002:** Targeted therapies for recurrent disease or disease progression.

Target	Prevalence	Therapy
IDH1	10–20%	Ivosidenib
FGFR2	9–15%	Futibatinib, Pemigatinib, Infigratinib
HER2	5–20%	Trastuzumab + Pertuzumab, Neratinib, Lapatinib
BRAF V600E	<5%	Dabrafenib + Trametinib
TMB-H	<5%	Nivolumab + Ipilimumab
MSI-H/dMMR	<5%	Pembrolizumab
NTRK	<1%	Entrectinib, Larotrectinib
RET	<1%	Selpercatinib, Pralsetinib

## Data Availability

No new data were created or analyzed in this study. Data sharing does not apply to this article.

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
