# Peer review of "Management of Intrahepatic Cholangiocarcinoma: A Narrative Review"

_cancers, 2024, doi:10.3390/cancers16040739_

Round 1
Reviewer 1 Report (Previous Reviewer 1)
Comments and Suggestions for Authors
This manuscript is a resubmission of review article on the management of intrahepatic cholangiocarcinoma. The authors addressed many of the points this reviewer outlined on previous review. This is resulted in significant improvement in the manuscript and includes important elements for the management of intrahepatic cholangiocarcinoma.
Author Response
Responses attached

Reviewer 2 Report (Previous Reviewer 2)
Comments and Suggestions for Authors
1. Explore the most recent findings and advancements in our knowledge of the molecular mechanisms regulating cholangiocarcinoma. Explore the genetic mutations, molecular alterations, and signaling pathways that are associated with the progression and development of CCA.
2. Present a comprehensive synopsis of the latest diagnostic approaches and instruments utilized in the early detection of cholangiocarcinoma.
3: Highlight the current clinical trials that are investigating novel therapies for cholangiocarcinoma. Reduce the effects of recent scientific discoveries regarding the advancement of forthcoming therapeutics.
4. Additionally, it is crucial to include the modifications that transpire throughout the progression of CCA in terms of their molecular mechanisms with an illustration.
Comments on the Quality of English Language
No any
Author Response
Thank you for re-reviewing our manuscript. We are pleased that our manuscript was favorably reviewed. We believe that the reviewers’ comments and the corresponding responses will serve only to strengthen the manuscript.
As requested, we have provided a point-by-point response to the referee comments below with changes to the manuscript highlighted in yellow.
Reviewer 2
- Explore the most recent findings and advancements in our knowledge of the molecular mechanisms regulating cholangiocarcinoma. Explore the genetic mutations, molecular alterations, and signaling pathways that are associated with the progression and development of CCA.
While we agree with the reviewer that there are exciting findings related to the molecular mechanisms regulating intrahepatic cholangiocarcinoma, a thorough review on this topic is well beyond the scope of this review. As requested by the editors, this review is focused on the management of intrahepatic cholangiocarcinoma for potentially resectable patients. As such, we have reviewed the available literature related to the surgical management of these patients and do not intend to perform a thorough systematic review of the entire topic.
- Present a comprehensive synopsis of the latest diagnostic approaches and instruments utilized in the early detection of cholangiocarcinoma.
We have provided a synopsis of the diagnostic approaches and instruments utilized in the detection of cholangiocarcinoma which can be found below. To our knowledge, there are no instruments currently utilized or approved for the screening or “early detection” of intrahepatic cholangiocarcinoma.
If suspected, a thorough history and physical examination should be the first step in the evaluation, with a focus on risk factors for chronic liver inflammation (e.g., primary sclerosing cholangitis, metabolic dysfunction-associated steatotic liver disease, alcoholic cirrhosis, viral hepatitis, autoimmune hepatitis), hepatotoxin exposure (thoratrast, aflatoxin), travel history (liver fluke infection), family history of liver malignancy, and personal colorectal cancer screening history. In addition to routine laboratory workup including liver function tests, the tumor markers carcinoembryonic antigen (CEA), carbohydrate antigen 19-9 (CA 19-9), and alpha-fetoprotein (AFP) should be included. While CA 19-9 is non-specific given elevations in inflammatory states as well as other cancers, it has been demonstrated to have a 72% sensitivity rate for ICC [12]. However, it should be noted that 5-10% of the Caucasian population does not synthesize CA 19-9 and are at risk for false negatives [13].
Imaging is crucial for diagnosis and assessment of resectability, with both multiphasic computed tomography (CT) or magnetic resonance imaging (MRI) with intravenous contrast as effective modalities. ICC can be distinguished from HCC as there is typically progressive contrast uptake in both arterial and venous phases of cross-sectional imaging, whereas HCC is more classically associated with washout during the venous phase [14]. Additional image findings that suggest ICC include liver capsule retraction and biliary dilation in the vicinity of the lesion. It is important to note that a third of patients may not have these classic findings on imaging [15]. A CT chest should be obtained as part of the cancer staging. An image-guided biopsy can be a useful adjunct if imaging is equivocal. If a biopsy reveals adenocarcinoma, an esophagogastroduodenoscopy (EGD) and colonoscopy should be performed to rule out a metastatic liver lesion. Comprehensive molecular profiling is recommended for patients with unresectable or metastatic disease who are candidates for systemic therapy [16].
3: Highlight the current clinical trials that are investigating novel therapies for cholangiocarcinoma. Reduce the effects of recent scientific discoveries regarding the advancement of forthcoming therapeutics.
We have added a comment regarding ongoing clinical trials related to the surgical management of intrahepatic cholangiocarcinoma. We apologize, but we are unclear as to the reviewer’s request to “reduce the effects” of recent discoveries regarding the advancement of forthcoming therapeutics.
Promising ongoing clinical trials for patients with potentially resectable ICC are largely focused on perioperative administration of immune-based, cytotoxic, or molecular-based personalized targeted therapies in patients undergoing curative-intent resection. Ongoing adjuvant trials following curative-intent resection of ICC include the ATTICA-1 trial (phase 3 randomized trial of gemcitabine/cisplatin vs. observation), the GAIN trial (phase 3 randomized trial of perioperative gemcitabine/cisplatin vs. surgery alone), and the OPTIC trial (phase II trial of neoadjuvant nab-paclitaxel, cisplatin and gemcitabine with or without infigratinib).
- Additionally, it is crucial to include the modifications that transpire throughout the progression of CCA in terms of their molecular mechanisms with an illustration.
Please see response 1 above as this is beyond the scope of this review for the management of patients with potentially resectable ICC.
Round 2
Reviewer 2 Report (Previous Reviewer 2)
Comments and Suggestions for Authors
All of my remarks have been satisfactorily included by the authors.
Comments on the Quality of English LanguageNo any
This manuscript is a resubmission of an earlier submission. The following is a list of the peer review reports and author responses from that submission.
Round 1
Reviewer 1 Report
Comments and Suggestions for Authors
This manuscript is a review article on the management of resectable intrahepatic cholangiocarcinoma. The authors provide a comprehensive review of the field. The article is well-written and easy to comprehend by readers.
While comprehensive, excessive time is spent on the discussion of systemic chemotherapy and advanced disease. While the authors couch this review as the management of “disease recurrence", seemingly as a reason to elaborate on systemic therapies. This article then should be entitled to include unresectable cholangiocarcinoma as well.
Table 1 is a simplistic summary of the recommendations of systemic therapy for ICC. This table is not useful. Consideration for a different table, perhaps specifying factors associated with resectability and/or high risk features (eg lymph node positivity on imaging, elevated CA 19-9, satellitosis). Another figure might consider more careful description of the nodal stations, instructing the readers regarding how to adequately perform a lymphadenectomy.
The authors should elaborate more on management of the resectable ICC patient, including the oncologically high risk. While resectability may be typically defined based on anatomic parameters, further discussion regarding whether patients who have grossly enlarged perihilar lymph nodes should be operated upon front. What about the patient with preoperative CA 19-9 >1000.? What about findings of satellitosis? Is there any role for not operating on such patients? If so, when and when not? For example, patients with extensive unilobar satellitosis be offered upfront surgery? What about patient's where the satellite lesions cross into the contralateral hemiliver? Is there any role for neoadjuvant therapy in these high risk patients?
The authors emphasize the importance of lymphadenectomy. While this reviewer concurs, further evidence should be presented regarding the benefit. Does the lymphadenectomy afford a therapeutic benefit? If not, and simply a guide for risk, how does the lymph node status guide adjuvant therapy? Are there data supporting differences in adjuvant therapy based on nodal status? Is the importance of lymphadenectomy simply for prognostic value?
The authors should mention only that “diagnostic laparoscopy should be considered". Further detail may be helpful to the reader regarding this. They only site one old study. Should all patients undergo staging laparoscopy? Only those with high oncologic risk factors?
The authors emphasize the relatively high rate of actionable mutations in these patients. However, little rationale is provided regarding how this information helps manage the resectable ICC. For example, should a patient with preop known FGFR2 mutation be offered upfront surgery, even with high risk features such as satellitosis or high CA 19-9? What is the role of targeted therapies in patients with actionable mutations determined postoperatively. The authors spent excessive time discussing features more relevant to patients with unresectable advanced disease rather than resectable at ICC.
It is understood that there are limited data on many of these important questions in the management of resectable ICC, perhaps explaining why the authors spend more time discussing the management of advanced/unresectable disease (ie recurrence). However, even with limited data, the authors should at least discuss and perhaps advise on these management challenges.
Reviewer 2 Report
Comments and Suggestions for Authors
The review paper titled "Management of Resectable Intrahepatic Cholangiocarcinoma: A Narrative Review" aims to provide a thorough examination of several facets related to the treatment and control of intrahepatic cholangiocarcinoma (ICC). Nevertheless, there are a few recommendations for improvement:
1- The paper shows an insufficient organizational framework, making it difficult for readers to figure out the ordered sequence of information. To enhance the understanding and navigation of the information, it is advisable to arrange it into parts with separate headings and subheadings.
2- To improve transparency and consistency, the article mentions a literature search in introducing without getting into comprehensiveness about the databases, keywords, or inclusion/exclusion criteria.
3- The discussion portion is concise and might be enhanced by a more thorough examination of the literature, encompassing an in-depth evaluation of the approach to research and limitations.
4- Tables and figures, which might improve the presentation of complicated material, are absent from the work. When presenting important data, treatment procedures, or research features graphically, think about using tables, figures, or flowcharts.
Comments on the Quality of English Language
The paper has several grammatical and typographical mistakes.